# Improving Soft Unification with Knowledge Graph Embedding Methods

## Abstract

Neural Theorem Provers (NTPs) present a promising framework for neuro-symbolic reasoning, combining end-to-end differentiability with the interpretability of symbolic logic programming. However, optimizing NTPs remains a significant challenge due to their complex objective landscape and gradient sparcity. On the other hand, Knowledge Graph Embedding (KGE) methods offer smooth optimization with well-defined learning objectives but often lack interpretability. In this work, we propose several strategies to integrate the strengths of NTPs and KGEs. By incorporating KGE objectives into the NTP framework, we demonstrate substantial improvements in both accuracy and computational efficiency.

## 1 Introduction

Deep Learning (DL) methods have recently achieved tremendous progress in various tasks such as language modeling (Touvron et al., 2023; Liu et al., 2023) and content generation Rombach et al. (2022); Kerbl et al. (2023). However, when compared with symbolic systems, they are still limited by the long-lasting problems of the lack of interpretation, out-of-domain generalizability and reasoning abilities.

To address the above challenges, the concept of Neuro-Symbolic AI (NeSy) has been proposed to integrate DL and symbolic AI into one end-to-end differentiable system. A popular approach for such integration is to embed discrete symbols into continuous vector space to enable end-to-end differentiability (Rocktäschel & Riedel, 2017; Minervini et al., 2019; Badreddine et al., 2022). Neural Theorem Prover (Rocktäschel & Riedel, 2017) (NTP) is a representative of such approach. It introduces the concept of soft unification during backward chaining process, where the unification operation is on the learnt embedding space instead of between discrete symbols. Subsequent works Greedy Neural Theorem Prover (GNTP) (Minervini et al., 2019) and Conditional Theorem Prover (CTP) (Minervini et al., 2020) implement top-$k$ rule retrieval and rule reformulation to improve NTP's scalability and performance.

Although NTP has been shown to be effective on various datasets, it is known to be hard to optimize (Rocktäschel & Riedel, 2017; Minervini et al., 2019; Maene & Raedt, 2023; de Jong & Sha, 2019). Specifically, as NTPs adopt the fuzzy min-max semiring for unification score aggregation, only a fraction of embedding parameters will receive gradient updates. The model optimization is thus heavily dependent on the initialization, and can get stuck in local minima (de Jong & Sha, 2019), resulting in under-explored and unregularized embedding space. DeepSoftLog (Maene & Raedt, 2023) addresses the above limitation by using differentiable probabilistic semiring instead of fuzzy semiring, along with other proposed properties to smooth out the back-propagation process. However, as it requires additional modules for knowledge compilation (Darwiche, 2011) and requires all possible proofs to be considered during training (as opposed to $k$-best approximation), it is intrinsically hard to scale to larger datasets.

On the other hand, Knowledge Graph Embedding (KGE)s (Bordes et al., 2013; Lin et al., 2015; Yang et al., 2015; Trouillon et al., 2016; Dettmers et al., 2018; Sun et al., 2019) are state-of-the-art (SOTA) methods for modeling Knowledge Graphs (KGs). KGEs learn mappings from symbols into their corresponding vector representations by maximizing scores for positive triplets while minimizing for the negatives, based on some predefined score functions. KGEs enjoy well-defined loss functions, smooth optimization process, and have shown SOTA performances on KG tasks such as

link prediction. However, as KGEs are purely sub-symbolic algorithms, they lack the interpretability as compared to NTPs.

Motivated by the complementary properties of NTPs and KGEs, we conduct the first systematic study for integrating KGEs into the NTP framework. The rest of the paper is arranged as follow: in Section 3 we provide brief definition and introduction for NTPs and KGEs, and discuss the hardness of training NTP from an embedding perspective 3.4; In Section 4 we explain four strategies for integrating KGEs with NTPs, and conduct detailed experiments in Section 5. Finally, we provide ablation studies to examine important components during the integration between KGEs and NTPs. We wish our work can serve as a first step to future studies on such integration and to improve upon existing differentiable provers.

**Contribution:**

1. We provide the first systematic study for integrating KGEs into NTPs and propose four integration strategies, with two focusing on performance, and the other two on efficiency.

2. We show that the integration noticeably improve the baseline NTP by a large margin and achieve SOTA results on multiple datasets. We also show that by leveraging the properties of KGEs we could drastically improve the inference and evaluation efficiency.

3. We provide detailed ablations to examine the learnt embedding of NTPs and key factors in the integration. Interestingly, we find NTPs can achieve superior results with pure KGE objectives under several datasets, suggesting the synergy between the two distinct methods.

## 2 RELATED WORK

**Differentiable Logic Programming** algorithms can be roughly divided into two categories. (1) Disentangled perception+reasoning (Manhaeve et al., 2018; Huang et al., 2021; Yang et al., 2023). This line of works train a neural network to output a probability distribution over symbols, which is then consumed by a differentiable logic solver. For example, DeepProbLog Manhaeve et al. (2018) guides a neural network with probabilistic circuits constructed by Sentential Decision Diagram (Darwiche, 2011) (SDD). Scallop (Huang et al., 2021) scales up DeepProbLog by only considering top-k possible worlds. NeurASP (Yang et al., 2023) adopts the same strategy, but replace SDD with a Answer Set Programming solver. Under this regime, the neural component is completely separated from the reasoning module. (2)Soft logic programming (Cohen, 2016; Badreddine et al., 2022; Yang et al., 2017; Rocktäschel & Riedel, 2017). This line of works are a continuous relaxation on top of logic programming, by learn a mapping from symbols and logic operations into latent embeddings and differentiable tensor operations. Logic Tensor Network (Badreddine et al., 2022) extends First-Order Logic (FOL) with fuzzy semantics. NEURALLP (Yang et al., 2017) is a rule-based learning algorithm that extends TensorLog (Cohen, 2016) by learning to soft select and compose rules. Besides, Neural Theorem Prover (Rocktäschel & Riedel, 2017) learns latent embeddings for symbols following backward chaining algorithm. Greedy Neural Theorem Prover (Minervini et al., 2019) and Conditional Theorem Prover (Minervini et al., 2020) improve the scalability of NTP by incorporating top-k retrieval and soft rule reformulation.

**Knowledge Graph Embedding** (KGE) are SOTA methods for link prediction tasks over large-scale KGs. TRANSE (Bordes et al., 2013) and its extensions (Wang et al., 2014; Xiao et al., 2015) are translation-based KGEs which minimize distance between subject and object, translated by the predicate. On the other hand, RESCAL (Nickel et al., 2011), COMPLEX (Trouillon et al., 2016), TUCKER (Balazevic et al., 2019) etc. use multi-linear maps to combine subject, relation and object for score calculation.

**Path-based KG Algorithms** explicitly learn the multi-hop paths over KGs. They can be applied directly on top of KGEs by handling multi-hop relation paths as compositions over embedding space such as in (Lin et al., 2015), or can be formulated as path-searching algorithms, optimized by Reinforcement Learning objectives such as in (Das et al., 2018; Zhu et al., 2023; Lin et al., 2018).

## 3 BACKGROUND

### 3.1 NEURAL THEOREM PROVER

In this section we define the syntax and briefly introduce the SLD resolution and NTP algorithm. We refer the reader to (Rocktäschel & Riedel, 2017) for a more in-depth explanation.

**Syntax** A term t can be either a constant $c$ or a variable $\mathbf{X}$[1]. An atom is defined as a combination of a predicate symbol and a list of terms. Rules are in the form of $\mathbf{H}$ :– $\mathbb{B}$, where the head $\mathbf{H}$ is an atom, and the body $\mathbb{B}$ is a list of atoms connected by conjunctions. A rule with no free variables is called a ground rule, and a ground rule with an empty body is called a fact. A substitution, denoted as $\psi = \{\mathbf{X}_1/t_1, \ldots, \mathbf{X}_N/t_N\}$ , assigns variables $\mathbf{X}_i$ to terms $t_i$, and applying a substitution to an atom replaces each occurrence of $\mathbf{X}_i$ with the corresponding term $t_i$. In this work, we only consider atoms with binary predicates in the form of $(s, r, o)$, where $s$, $r$ and $o$ denote subject, predicate (relation) and object respectively.

**Backward Chaining** Given the goal, backward chaining works backward to find supporting facts and rules from the Knowledge Base (KB). It can be seen as an iterative process of applying AND/OR: the OR operation looks for all rules with matching head to perform unification. The AND module is subsequently called to iteratively prove all atoms in the unified rule's body, where the OR module is again called recursively.

**NTP and Soft Unification** NTPs provide a continuous relaxation of backward chaining by introducing *soft unification*. It calculates a unification score $\gamma = \phi_{\text{NTP}}(c_i, c_j)$ over the embeddings of two symbols, where $\phi_{\text{NTP}}$ refers to the predefined similarity function, $c_i$ and $c_j$ denotes two constant terms to be unified. In case of NTP, a Gaussian kernel is usually adopted for $\phi_{\text{NTP}}$. The unification score $\gamma$ at each proof state are then aggregated following the min/max fuzzy semiring, also known as the Gödel $t$-norm. Specifically, the AND module performs min aggregation as all sub-goals have to be proved for the given rule, and OR perform max aggregation, since we only need one proof to be true to prove the goal. During training, given a KG $\mathcal{G}$, each fact $(s, r, o) \in \mathcal{G}$ is corrupted to obtain negative samples $(s', r, o)$, $(s, r, o')$ and $(s', r, o') \notin \mathcal{G}$. The learning objective is then defined as the negative log likelihood of the aggregated unification score:

$$\mathcal{L}\text{NTP}_\theta^{\mathcal{G}} = \sum_{((s,r,o),y)\,\in\,\mathcal{G}} -y \log(\text{NTP}_\theta^{\mathcal{G}}((s,r,o)) - (1-y)\log(1 - \text{NTP}_\theta^{\mathcal{G}}((s,r,o)) \tag{1}$$

where $\text{NTP}_\theta^{\mathcal{G}}$ denotes the NTP module with KG $\mathcal{G}$, parameterized by $\theta$.

### 3.2 KGS AND EMBEDDING METHODS

A Knowledge Graph (KG) $\mathcal{G}$ is a directed multi-graph, represented as a collection of triplets (facts) $(s, r, o) \subseteq \mathcal{E} \times \mathcal{R} \times \mathcal{E}$, where $\mathcal{E}$ and $\mathcal{R}$ denote the set of entities and relations in $\mathcal{G}$. A KGE model defines a function that maps triplets to scores $\phi_{\text{KGE}} : \mathcal{E} \times \mathcal{R} \times \mathcal{E} \to \mathbb{R}$. This score function $\phi_{\text{KGE}}$ can be translation-based as in TRANSE (Bordes et al., 2013): $\phi_{\text{TRANSE}}(\boldsymbol{s}, \boldsymbol{r}, \boldsymbol{o}) = -||\boldsymbol{s} + \boldsymbol{r} - \boldsymbol{o}||$, or similarity-based using a multi-linear function (Trouillon et al., 2016; Yang et al., 2015). For instance, COMPLEX (Trouillon et al., 2016) defines the score function as $\phi_{\text{COMPLEX}} = \text{Re}(\langle \boldsymbol{s}, \boldsymbol{r}, \overline{\boldsymbol{o}} \rangle)$, where $\langle \cdot, \cdot, \cdot \rangle$ denotes the tri-linear product, $\text{Re}$ denotes the real part of the complex number, and $\overline{\cdot}$ denotes the complex conjugate. KGEs are traditionally interpreted as energy-based models (EBMs), where the score is interpreted as the negative energy of triplets, and are trained with contrastive objectives and negative log likelihood loss, similar to $\mathcal{L}_{\text{NTP}}$. Besides treating KGEs as EBMs, existing works (Joulin et al., 2017; Lacroix et al., 2018; Ruffinelli et al., 2020) have shown that KGEs can be effectively trained using cross-entropy loss to predict missing object over $\mathcal{E}$, given subjects and predicates, *i.e.* by maximizing:

$$\log p(o \mid s, r) = \phi_{\text{KGE}}(s, r, o) - \log \sum_{o' \in \mathcal{E}} \exp \phi_{\text{KGE}}(s, r, o') \tag{2}$$

---

[1]We focus on function-free First Order Logic, and therefore does not consider structured terms.

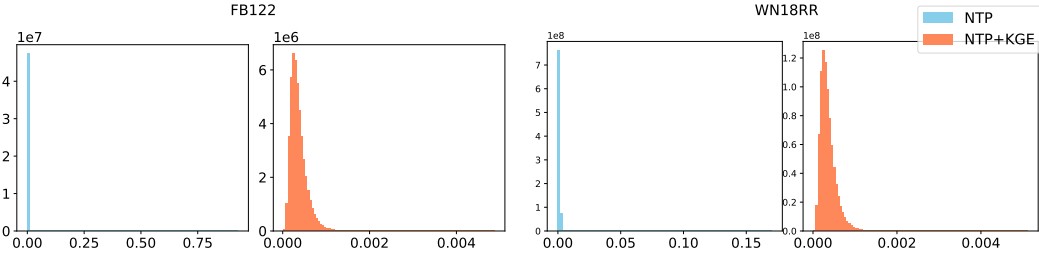

Figure 1: Illustration of CTP algorithm with a transitive rule template and depth $= 1$. Given a goal $(s, R, o)$, it first transforms the goal predicate to a list of predicates forming the proof path. Then it takes the known subject $s$ and predicate $R_i$ to predict the latent object $z_i$ with top-$k$ retrieval; it then uses the predicted $z_i$ as the next subject and predict $z_{i+1}$ to step through the proof path.

### 3.3 NTPs as Memory-Augmented Path-based Algorithm

Inspired by Conditional Theorem Prover (CTP) (Minervini et al., 2020) we can implement NTP as a memory-augmented path-based algorithm. Instead of searching for all rules in the KB, CTP extends NTP by learning a goal transformation module that directly transforms each goal predicate to a list of predicates following pre-set rule templates (*e.g.* transitivity), thereby forming the proof paths. Given a (sub)goal, the model steps through each atom formed by the transformed goal predicate until it reaches the end of the path. The above procedure is instantiated recursively for each atom (sub-goal) along the path until it reaches the depth limit. This formulation gives us more flexibility for integrating KGE methods comparing to original NTP. In Figure 1 we show a simple example of CTP with $depth = 1$ and one transitive rule template of length $n$. At each step, the process can be viewed as sampling $k$ plausible objects given the subject and predicate $o \sim \mathcal{P}(s, r)$, which shares similar formulation as in formula 2.

### 3.4 Hardness in Training NTPs

Previous works (Rocktäschel & Riedel, 2017; Maene & De Raedt, 2023; de Jong & Sha, 2019) have primarily focused on analyzing and addressing the limitations of NTPs from the perspective of unsmooth optimization, particularly in relation to the sparse gradient problem. However, attempts to mitigate this issue often introduce additional computational overhead. For example, DeepSoft-Log (Maene & De Raedt, 2023) tackles the sparse gradient problem in NTP training by employing differentiable probabilistic semantics, combined with a knowledge compilation step for probabilistic inference, and evaluates the entire proof tree (as opposed to using a top-$k$ approximation) to ensure accurate gradient calculation. While this approach yields improved accuracy and provides a more interpretable probabilistic framework, it struggles to scale beyond small KBs.

Figure 2: Distribution of pairwise similarity from CTP (blue) and CTP combined with KGE (orange).

In contrast to previous works, we try to view the hardness in NTP training from the embedding perspective. Unlike KGEs, which compute triplet scores based on interactions between entities and predicates, NTPs derive embeddings solely from pairwise unification scores. This results in embeddings in NTPs being less structured. Furthermore, while KGEs typically sample a large number of negative examples (*e.g.*, 256) to learn the distribution of entities given a subject/object and relation: $(o \sim \mathcal{P}(s, r))$ or $(s \sim \mathcal{P}(r, o))$, NTPs generally sample only a single negative example per entity and retrieve only the top-$k$ facts from the KB for unification, where $k \ll |\mathcal{E}|$. As a result, semantically similar embeddings in NTPs may end up in vastly different regions of the embedding space if they are never unified or do not receive gradient updates due to the fuzzy min/max operations. In Figure 2 we show the distribution of pairwise unification scores between entities, and we could observe the pairwise score distribution for CTP (blue) is mostly close to 0, suggesting only a handful of embed-

| Modules | CTP | | VARIANTS |
|---|---|---|---|
| step | $i = \text{topk}^{\mathcal{G}}(s, r), k); z = \mathcal{G}[i][-1]$ | CTP$_3$: | $z = \text{trans}(s, r); i = \text{topk}^{\mathcal{G}}((s, r, z), k)$ |
| score$_{\text{latent}}$ | $\gamma = \phi_{\text{NTP}}((s, r, z), \mathcal{G}[i])$ | CTP$_2$: | $\gamma = \lambda\phi_{\text{NTP}}((s, r, z), \mathcal{G}[i]) + (1 - \lambda)\phi_{\text{KGE}}(s, r, z)$ |
| score$_{\text{final}}$ | $i = \text{topk}^{\mathcal{G}}((z, r, o), k); \gamma = \phi_{\text{NTP}}((z, r, o), \mathcal{G}[i])$ | CTP$_4$: | $\gamma = \phi_{\text{KGE}}(z, r, o)$ |
| loss | $\mathcal{L} = \mathcal{L}_{\text{NTP}}$ | CTP$_1$: | $\mathcal{L} = \lambda\mathcal{L}_{\text{NTP}} + (1 - \lambda)\mathcal{L}_{\text{KGE}}$ |

Table 1: Summary of the proposed four variants for integration. We consider four modules in CTP to inject KGEs: 1) **step**: Given $(s, r)$ find $o$; 2) **score**$_{\text{latent}}$: unification score along each proof path; 3) **score**$_{\text{final}}$: unification score calculation at the last proof step, and 4) **loss**: the final loss calculation. Column CTP shows the original CTP algorithm, and Column VARIANT shows the modified algorithm by integrating KGEs with the corresponding modules. The variant only differs from the original CTP for the corresponding module (for instance, CTP$_1$ includes the KGE objective in the final loss function. This is the only difference between baseline CTP and CTP$_1$, and all other variants do not include the KGE objective in their loss function). $\mathcal{G}$ denotes the KG, and $\mathcal{G}[i]$ refers to the $i$-th facts in the KG. trans denotes the translation function of KGEs, $z$ refers to the tail entity predicted by $(s, r)$, and $\text{topk}^{\mathcal{G}}$ denotes the top-$k$ retrieval from $\mathcal{G}$ that returns the top-$k$ indices $i$.

ded symbols have interactions with each other. This lack of interaction can lead to an unstructured and suboptimal embedding space, negatively affecting the performance of NTPs.

Therefore, given the above challenge in training NTPs, in this work we explore different strategies for leveraging the strengths of KGEs to regularize and enhance the embedding space of NTPs, given the proven effectiveness of KGEs in learning structured representations.

## 4 METHOD

In this section we discuss the four variants we considered for integrating KGEs with NTPs. In Table 1 we summarize how each variant are implemented on top of the original CTP framework.

**KGE as an auxiliary loss model** The most straightforward strategy for leveraging KGEs to support NTP training is to use KGE as an auxiliary model for loss calculation. The overall loss for training NTPs then becomes:

$$\mathcal{L} = \lambda\mathcal{L}_{\text{NTP}_\theta^{\mathcal{G}}} + (1 - \lambda)\sum_{((s,r,o),y) \in \mathcal{G}} -y\log(\text{KGE}_\theta((s, r, o))) - (1 - y)\log(1 - \text{KGE}_\theta((s, r, o)))$$

where $\lambda$ is a hyper-parameter controlling the weight for the mixture. We denote this variant as CTP$_1$. Note that using KGE as an auxiliary loss term was briefly mentioned in the original NTP paper (Rocktäschel & Riedel, 2017). However, it was not further examined nor was it used in the subsequent works in GNTP and CTP.

**KGE as an auxiliary score function** Similar to CTP$_1$, we again consider utilizing KGE score function. But rather than appending it as a loss term at the very end, here we inject KGE score function $\phi_{\text{KGE}}$ into NTPs as an auxiliary score $\phi_{mixed} = \lambda\phi_{\text{NTP}} + (1 - \lambda)\phi_{\text{KGE}}$. In this way, we could provide additional regularization at each proof step, and force the model to learn interactions between entities and predicates *along* the proof path. We refer to this variant as CTP$_2$. Despite the simplicity, we find this variant to bring the most consistent improvement across most experiments.

**KGE for stepping through** For translation-based KGEs such as TRANSE and ROTATE, the tail object $o$ can be efficiently calculated given $(s, r)$. To leverage this translational property, we consider replacing the topk retrieval with a translation-based operation to improve inference efficiency. Specifically, given a $(s, r)$ pair, we use translational KGE to obtain corresponding object and then retrieve the closest $k$ facts for score calculation. During inference we skip the retrieval and score calculation. This variant, referred to as CTP$_3$, is designed to improve the efficiency of NTPs. In this case, for each proof path, CTP$_3$ is very similar to the path-based KGE method PTRANSE (Lin et al., 2015). However, they differ in that 1) PTRANSE follow KGE training strategy, and utilize additional prior for spurious relations, while 2) CTP$_3$ calculates unification scores along the proof path, and uses the original NTP's retrieval-based score calculation for each proof path.

**KGE for final score calculation** We consider applying KGE at the final step at each proof path. One drawback on NTPs' efficiency is their evaluation speed. During evaluation of a link prediction task, in order to rank all the entities in the KG, the model retrieves top-$k$ facts for each combination of the missing entity and the known predicate-object/subject pair, followed by the unification score calculation between two tensors of shape $(|\mathcal{E}|, k, 3d)$ where $k$ is the retrieved $k$ facts, and $d$ is the embedding dimension. For example, WN18RR dataset contains 40,943 entities. With $k = 10$ we need to compute the pairwise distance with the Gaussian kernel between two matrices of shape $(40,943 \times 10, \ 3d)$. This is done at the end of *every* proof path, leading to extremely slow evaluation compared to KGEs. Therefore, we try to replace the last proof step with KGEs, while keeping the previous steps with NTP. In this way, we wish to leverage the multi-hop reasoning ability of NTPs while using KGEs for local ranking at the final step. We refer to this variant as $CTP_4$.

While it is trivial to combine any variants together, we do not find performance gain by doing so. Therefore we leave them separated for the sake of clarity.

## 5 EXPERIMENTS

### 5.1 EXPERIMENTAL SETUPS

**Dataset** We conduct experiments on popular link prediction datasets including Nations, UMLS and Kinship (Kemp et al., 2006). Following GNTP (Minervini et al., 2019) we also experiment on FB122 (Guo et al., 2016a) and WN18RR (Dettmers et al., 2018). FB122 consists of two test split: Test-I and Test-II, where Test-II contains the set of triplets that can be inferred via logic rules, and Test-I denotes the other triplets. We follow the same evaluation protocol as in GNTP and CTP, and report Mean Reciprocal Rank (MRR) and HITS@$m$ under the filtered setting.

**Baseline** We compare our work with the previous NTPs: GNTP and CTP. Following GNTP, we also compare with additional neuro-symbolic systems: NEURALLP (Yang et al., 2017) and MINERVA (Das et al., 2018) on Kinship, Nations and UMLS datasets. NEURALLP extends on TENSORLOG (Cohen, 2016) by also learning rules; MINERVA deploys REINFORCE algorithm. For FB122 and WN18RR, we also compare against popular KGE methods COMPLEX and DISTMULT.

**Implementation** We conduct our experiments primarily on CTP (Minervini et al., 2020). Since the original CTP did not evaluate on large-scale dataset FB122 and WN18RR, we perform hyper parameter tuning to obtain the CTP baseline for these two datasets. By default, we use COMPLEX for $CTP_1$, $CTP_3$ and $CTP_4$, and ROTATE for $CTP_2$ as we observe best overall performance under these settings. During training, we obtain negative samples by corrupting subject, entity, and both, each with $n$ times, resulting in $3n$ negative samples generated for each triplet. These negative samples will receive negative label $y = 0$, and the model is trained according to the NTP objective (Eq. 1). For $CTP_1$ and $CTP_2$ we use $\lambda = 0.5$ as the default weight for combining KGE and NTP.

### 5.2 RESULTS

**Nations, Kinship and UMLS** In Table 2 we show link prediction results on Kinship, Nation and UMLS datasets. We can observe that $CTP_2$ consistently outperforms CTP by a large margin, and achieve SOTA results on Nations and UMLS datasets. For instance, on Nation and UMLS datasets, $CTP_2$ achieve 0.788 MRR and 0.851 MRR respectively, as comparing to 0.709 MRR and 0.80 MRR for CTP. On the other hand, $CTP_1$ achieves best results on the Kinship dataset with 0.75 MRR and surpasses baseline CTP on UMLS dataset, but lags behind on the Nations dataset.

**FB122 and WN18RR** In Figure 3 we show validation MRR during training on FB122 dataset for baseline CTP (blue), $CTP_2$ with COMPLEX (green) and DISTMULT (orange). We can observe that both $CTP_2$ converges quickly in the first 20 epochs, with $CTP_2$-COMPLEX slightly higher than $CTP_2$-DISTMULT, and both have much higher accuracy than the baseline CTP. In Table 3 and Table 4, we show link prediction results on the FB122 and WN18RR dataset. In FB122 dataset we can again observe that $CTP_2$ noticeably improve baseline CTP. Under all the models without access to the ground-truth rules, $CTP_2$ achieves best results under Test-II and Test-ALL splits with 0.681 MRR, 0.04 higher than the 2nd highest. Notably, while $CTP_1$ outperforms the CTP baseline on

| Datasets | Metrics | $CTP_1$ | $CTP_2$ | $CTP_3$ | $CTP_4$ | NTP | GNTP | CTP | NEURALLP | MINERVA |
|---|---|---|---|---|---|---|---|---|---|---|
| Kinship | MRR | **0.75** | 0.71 | 0.51 | 0.59 | 0.35 | 0.719 | 0.71 | 0.62 | 0.72 |
| | HITS@1 | **61.59** | 57.49 | 49.18 | 48.92 | 24 | 58.6 | 56.5 | 47.5 | 60.5 |
| | HITS@3 | **85.01** | 82.44 | 71.47 | 67.94 | 37 | 81.5 | 82.6 | 70.7 | 81.2 |
| | HITS@10 | **95.95** | 95.61 | 92.84 | 90.13 | 57 | 95.8 | 95.3 | 91.2 | 92.4 |
| Nations | MRR | 0.63 | **0.79** | 0.53 | 0.55 | 0.61 | 0.658 | 0.71 | - | - |
| | HITS@1 | 44.36 | **68.93** | 31.84 | 34.26 | 45 | 49.3 | 56.2 | - | - |
| | HITS@3 | 77.64 | **85.62** | 51.92 | 52.84 | 73 | 78.1 | 81.3 | - | - |
| | HITS@10 | 98.86 | **99.70** | 83.06 | 79.48 | 87 | 98.5 | 99.5 | - | - |
| UMLS | MRR | 0.82 | **0.85** | 0.65 | 0.76 | 0.80 | 0.84 | 0.81 | 0.78 | 0.82 |
| | HITS@1 | 69.90 | **75.20** | 54.62 | 62.75 | 79 | 73.2 | 69.4 | 64.3 | 72.8 |
| | HITS@3 | 93.19 | **94.64** | 77.40 | 84.37 | 88 | 94.1 | 89.8 | 86.9 | 90 |
| | HITS@10 | 98.71 | **98.21** | 92.58 | 92.18 | 95 | 98.6 | 95.3 | 96.2 | 96.8 |

Table 2: Link prediction results on Kinship, Nations and UMLS datasets. HITS@$m$ are reported as %.

| | | Test-I | | | | Test-II | | | | Test-ALL | | | |
|---|---|---|---|---|---|---|---|---|---|---|---|---|---|
| | | H@3 | H@5 | H@10 | MRR | H@3 | H@5 | H@10 | MRR | H@3 | H@5 | H@10 | MRR |
| With Rules | $KALE_P$ | **38.4** | **44.7** | **52.2** | 0.32 | 79.7 | 84.1 | 89.6 | 0.68 | 61.2 | 66.4 | 72.8 | 0.52 |
| | $KALE_J$ | 36.3 | 40.30 | 44.90 | **0.33** | 98.0 | 99.0 | 99.2 | 0.948 | 70.7 | 73.1 | 75.2 | 0.67 |
| | $ASR_D$ | 37.3 | 41.0 | 45.9 | 0.33 | **99.2** | **99.30** | **99.4** | **0.984** | 71.7 | 73.6 | 75.7 | 0.67 |
| | KBLRN | - | - | - | - | - | - | - | - | **74.0** | **77.0** | **79.7** | **0.70** |
| Without Rules | TransE | 36.0 | 41.5 | 48.1 | 0.29 | 77.5 | 82.8 | 88.4 | 0.63 | 58.9 | 64.20 | 70.2 | 0.48 |
| | DistMult | 36.0 | 40.3 | 45.3 | 0.31 | 92.3 | 93.8 | 94.7 | 0.874 | 67.4 | 70.2 | 72.9 | 0.63 |
| | ComplEx | **37.0** | **41.3** | **46.2** | **0.33** | 91.4 | 91.9 | 92.4 | 0.887 | 67.3 | 69.5 | 0.72 | 0.64 |
| | GNTP | 28.6 | 31.2 | 35.8 | 0.28 | 94.2 | 95.8 | 96.0 | 0.92 | 61.5 | 63.2 | 64.5 | 0.61 |
| | CTP | 31.2 | 34.7 | 39.51 | 0.30 | 96.1 | 97.0 | 97.9 | 0.94 | 64.5 | 65.1 | 68.3 | 0.63 |
| | $CTP_1$ | 30.6 | 33.1 | 37.8 | 0.29 | 95.0 | 95.9 | 96.6 | 0.89 | 60.4 | 61.3 | 62.9 | 0.56 |
| | $CTP_2$ | 34.4 | 38.2 | 43.1 | 0.32 | **99.1** | **99.2** | **99.4** | **0.98** | **69.9** | **71.32** | **73.0** | **0.68** |
| | $CTP_3$ | 25.3 | 30.2 | 34.2 | 0.25 | 93.7 | 94.5 | 94.8 | 0.83 | 59.4 | 60.8 | 62.2 | 0.53 |
| | $CTP_4$ | 30.2 | 32.7 | 37.1 | 0.28 | 94.5 | 95.4 | 95.9 | 0.85 | 61.1 | 64.6 | 67.4 | 0.61 |

Table 3: Link prediction result on FB122 dataset. Following GNTP (Minervini et al., 2019) we report accuracy on Test-I, Test-II and Test-ALL. H@$m$ are reported as %. $KALE_P$ and $KALE_J$ denote KALE-Pre and KALE-Joint from (Guo et al., 2016b). $ASR_D$ denotes ASR-DistMult from (Minervini et al., 2017). All the aforementioned models have access to the ground-truth logic rules, while other models in the table do not.

Kinship and UMLS datasets, we observe its performance to degrade on FB122 and WN18RR. An explanation could be that KGE models as EBMs generally require large amount of negative samples especially with large datasets. Therefore, given small amount of negatives, $CTP_1$ could work well on small datasets like Kinship and UMLS, but cannot scale to larger KBs like FB122 or WN18RR.

In all the experiments, we observe $CTP_2$ constantly outperform baseline CTP and other CTP variants in most but one dataset (Kinship), where $CTP_1$ achieves 0.75 MRR. This is because the length of proof paths in Kinship is always $\leq 1$, and therefore $CTP_2$ has little effect. We conjecture that the advantage of $CTP_2$ over $CTP_1$ is because it is injected into the NTP framework and regularize each latent subject predicted by the model along the proof path. Therefore it can be more effective at regularizing the embedding space comparing to appending the loss outside the proving process as in $CTP_1$. Moreover, as KGEs are usually trained with large numbers of negatives, directly adding KGE to the loss term of NTP may not be ideal. On the other hand, we observe performance degrades with $CTP_3$ and $CTP_4$. This is, however, expected. $CTP_3$ uses the

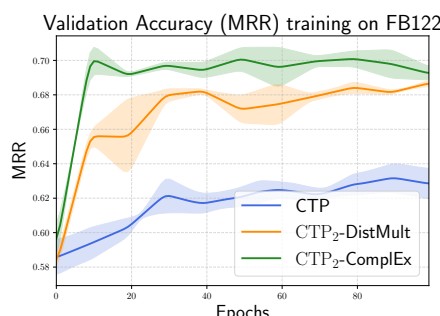

Figure 3: Validation MRR on FB122 dataset with baseline CTP and $CTP_2$ with DISTMULT and COMPLEX as integrated KGEs.

| Metrics | $CTP_1$ | $CTP_2$ | $CTP_3$ | $CTP_4$ | GNTP | CTP | COMPLEX | DISTMULT | NEURALLP | MINERVA |
|---------|---------|---------|---------|---------|------|-----|---------|----------|----------|---------|
| MRR | 0.361 | 0.425 | 0.326 | 0.304 | 0.381 | 0.364 | 0.415 | **0.463** | **0.463** | 0.448 |
| H@1 | 35.11 | 40.86 | 31.46 | 29.58 | 37.12 | 36.16 | 38.2 | **41.0** | 37.6 | 41.3 |
| H@3 | 36.05 | 43.50 | 33.57 | 33.42 | 38.54 | 36.72 | 43.3 | 44.1 | **46.8** | 45.6 |
| H@10 | 37.82 | 48.37 | 36.79 | 35.74 | 39.52 | 38.15 | 48.0 | **65.7** | **65.7** | 51.3 |

Table 4: Link prediction results on WN18RR. H@$m$ is reported as %.

translational property of TransE and RotatE to calculate the latent subject, which is equivalent to *top*-1 retrieval and can heavily suffer from spurious relation, as mentioned in Lin et al. (2015).

**Boosting NTP speed with KGE** Despite having lower accuracy, $CTP_3$ and $CTP_4$ can significantly improve the efficiency of inference and evaluation of NTP, especially on large-scale dataset. In Figure 4, we show per-sample inference and evaluation time under baseline CTP, $CTP_3$ and $CTP_4$. For inference, $CTP_3$ requires 2× and 7× less time compared to CTP on FB122 and WN18RR dataset, while $CTP_4$ reduces even further by 28× and 92×. For evaluation, $CTP_3$ requires 2× less time than CTP on both datasets, while $CTP_4$ reduces 942× and 1452× on FB122 and WN18RR.

## 5.3 ABLATION STUDIES

**Effect of using different KGEs** In Table 5 we show the performance of $CTP_1$, $CTP_2$ and $CTP_4$ using different KGE methods: ComplEx, DistMult, TransE and RotatE, and $CTP_3$ with TransE and RotatE. With $CTP_1$ and $CTP_2$, we can observe that the two similarity-based KGEs, COMPLEX DISTMULT generally yields the best performance, whereas translation-based KGE TransE and RotatE often lag back by a large margin. For instance, $CTP_1$ achieves 0.81 MRR on UMLS with COMPLEX, but only

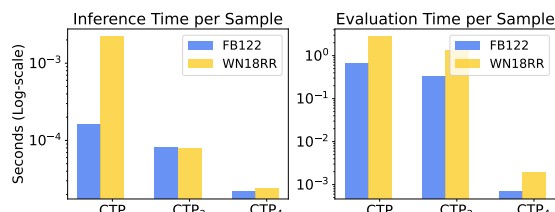

Figure 4: Second/sample on FB122 and WN18RR dataset on a NVIDIA V100 GPU with batch size = 512.

0.67 MRR under ROTATE. In general, we observe that $CTP_2$ is mostly invariant to the choice of KGE methods, followed by $CTP_1$, whereas $CTP_3$ and $CTP_4$'s performance can vary largely with different KGE methods. This is expected, as $CTP_1$ and $CTP_2$ are using KGE score functions as a regularization term, whereas $CTP_3$ and $CTP_4$ are making prediction directly based on KGEs.

**Regularized Embedding space** In Figure 5 we show the $t$-SNE visualization of the embedding space of original CTP and $CTP_2$-COMPLEX. For both methods, we could observe a few points being close to each other, suggesting the model are able to learn that they are *unifiable*. However, we can clearly observe $CTP_2$-COMPLEX also exhibits better global structures, whereas for CTP there only exists extremely local (pairwise) pattern. On the other hand, as shown in Figure 2, while baseline CTP (left, blue) exhibits extremely sparse connections between entities with unification score all gathered around 0, CTP combined with KGE objectives (right, orange) shows a much smoother score distribution, suggesting a much denser connectivity.

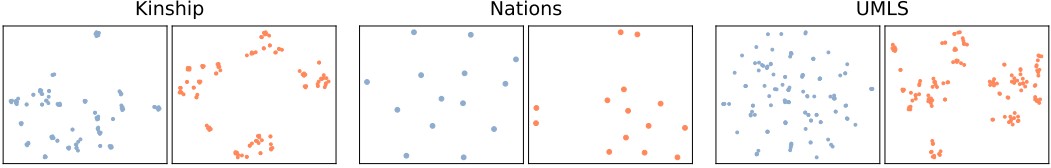

Figure 5: $t$-SNE visualization of embeddings for CTP (blue) and $CTP_2$ (orange) with perplexity = 5.

**Effect of weight $\lambda$ for combining KGE and NTPs** We find that the weight $\lambda$ controlling the combination of NTP and KGE loss/score function as in $CTP_1$ and $CTP_2$ plays an important rule

| | KGE | UMLS | | | | Kinship | | | | FB122 Test-ALL | | | |
|---|---|---|---|---|---|---|---|---|---|---|---|---|---|
| | | MRR | H@1 | H@3 | H@10 | MRR | H@1 | H@3 | H@10 | MRR | H@3 | H@5 | H@10 |
| CTP | - | 0.80 | 69.4 | 89.8 | 95.3 | 0.70 | 56.56 | 82.64 | 0.95 | 0.63 | 64.5 | 65.1 | 68.3 |
| $CTP_1$ | DistMult | 0.78 | 67.4 | 87.4 | 93.2 | 0.71 | 58.5 | 81.2 | 0.94 | 0.54 | 59.31 | **62.23** | **63.14** |
| | ComplEx | **0.81** | **68.9** | **93.1** | **98.7** | **0.74** | **61.6** | **85.0** | **95.94** | **0.56** | **60.4** | 61.3 | 62.9 |
| | TransE | 0.74 | 61.1 | 83.9 | 96.0 | 0.43 | 32.3 | 47.5 | 64.15 | 0.51 | 57.42 | 60.04 | 62.43 |
| | RotatE | 0.67 | 53.1 | 77.5 | 92.6 | 0.61 | 46.6 | 68.9 | 91.52 | 0.50 | 57.34 | 61.47 | 62.85 |
| $CTP_2$ | DistMult | 0.84 | 74.5 | 93.1 | 98.3 | 0.71 | 59.0 | 79.9 | 93.5 | 0.68 | 69.35 | 72.1 | **73.4** |
| | ComplEx | **0.85** | **75.2** | **94.6** | 98.2 | **0.72** | 57.0 | 82.3 | **95.3** | **0.68** | **69.9** | **71.3** | 73.0 |
| | TransE | 0.83 | 72.1 | 93.3 | 97.0 | 0.71 | 58.7 | **80.6** | 93.9 | 0.64 | 64.1 | 67.4 | 68.2 |
| | RotatE | 0.82 | 70.6 | 93.4 | 98.1 | 0.71 | **59.2** | **80.6** | 93.8 | 0.64 | 65.1 | 68.2 | 69.8 |
| $CTP_3$ | TransE | 0.48 | 36.6 | 57.4 | 78.2 | 0.49 | 40.3 | 68.12 | 90.54 | 0.31 | 28.8 | 35.7 | 44.4 |
| | RotatE | **0.65** | **54.6** | **77.4** | **92.5** | **0.54** | **45.2** | **71.47** | **92.84** | **0.53** | **59.4** | **60.8** | **62.2** |
| $CTP_4$ | DistMult | 0.72 | 57.1 | 78.2 | 89.0 | **0.61** | **49.7** | **69.52** | **91.7** | **0.62** | **62.4** | 64.32 | 66.8 |
| | ComplEx | **0.76** | **62.7** | **84.3** | **92.1** | 0.59 | 48.9 | 67.9 | 90.1 | 0.61 | 61.1 | **64.6** | **67.4** |
| | TransE | 0.58 | 50.3 | 72.4 | 90.1 | 0.53 | 44.6 | 63.21 | 90.5 | 0.48 | 55.3 | 57.8 | 59.0 |
| | RotatE | 0.61 | 49.5 | 74.2 | 91.8 | 0.50 | 43.9 | 63.7 | 89.2 | 0.60 | 61.4 | 64.0 | 67.0 |

Table 5: Link prediction results on UMLS, Kinship and FB122 dataset with different KGE models. Bold denotes the highest score for each variant under different KGE methods.

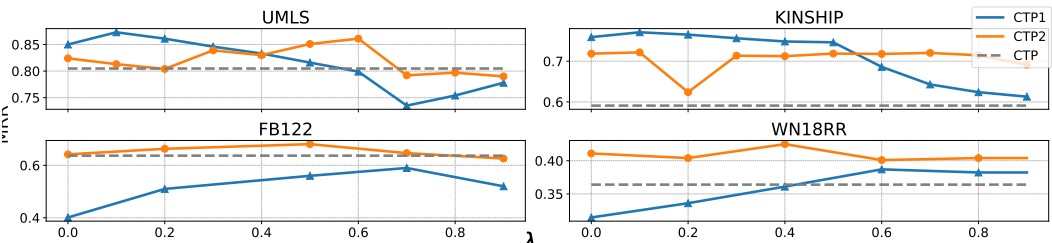

Figure 6: MRR with different weight $\lambda$ for combining NTP and KGE objectives.

on their performance. Therefore, we repeat experiments with different $\lambda$ on the tested datasets, and show the results in Figure 6. We can observe that performance of $CTP_1$ tends to fluctuate when $\lambda$ changes, while $CTP_2$ is more invariant. Specifically, on UMLS and Kinship dataset, the performance of $CTP_1$ increases with smaller $\lambda$; on the other hand, on FB122 and WN18RR the performance of $CTP_1$ increases with larger $\lambda$. Besides, for $CTP_1$ we find the training loss tends to be much more stable with smaller $\lambda$ as shown in Figure 2, which is as expected as the non-differentiable operations in CTP is smoothed out by the differentiable KGE loss calculation.

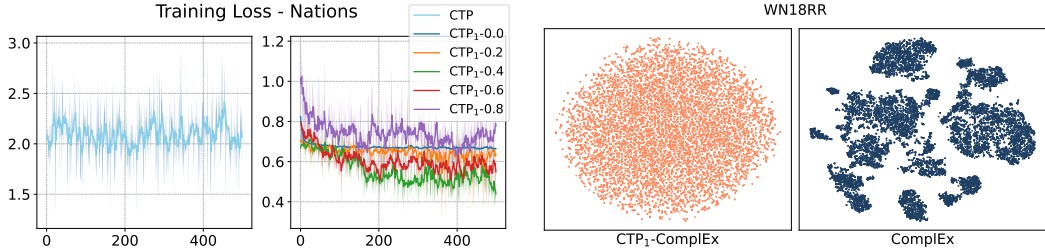

Figure 7: Training loss on Nations with CTP (left, blue) and $CTP_2$-COMPLEX (right) with different $\lambda$.

Figure 8: $t$-SNE visualization of entity embeddings from $CTP_2$-COMPLEX (left) and COMPLEX (right).

Interestingly, we find that both $CTP_1$ and $CTP_2$ maintain decent performance when $\lambda = 0$, suggesting both model perform well even when the loss/score is fully substituted by the KGE loss/score. For example, $CTP_1$ achieves SOTA performance of 0.87 MRR on UMLS when $\lambda = 0.1$, and 0.85 with $\lambda = 0$. The exception here is $CTP_1$ on FB122 and WN18RR dataset, where its performance decreases noticeably when $\lambda = 0$. One possible explanation is due to the missing KGE-specific

| $n$ | Metrics | CTP | CTP$_1$ | | | CTP$_2$ | | | CTP$_3$ | CTP$_4$ |
| | | | $\lambda = 0$ | $\lambda = 0.5$ | $\lambda = 0.8$ | $\lambda = 0$ | $\lambda = 0.5$ | $\lambda = 0.8$ | | |
|---|---|---|---|---|---|---|---|---|---|---|
| 1 | MRR | **0.64** | 0.40 | 0.56 | **0.59** | 0.64 | **0.68** | 0.65 | 0.53 | 0.32 |
| | HIT@3 | **64.50** | 46.24 | 60.40 | **62.43** | 0.65 | **69.43** | 65.83 | 59.40 | 34.80 |
| | HIT@10 | **68.30** | 50.07 | 62.90 | **63.75** | 67.50 | **73.01** | 68.76 | 62.20 | 41.52 |
| 16 | MRR | 0.61 | 0.43 | **0.59** | 0.57 | 0.63 | 0.65 | 0.49 | **0.55** | 0.45 |
| | HIT@3 | 62.50 | 47.62 | **60.17** | 59.49 | 64.25 | 67.03 | 50.03 | **60.84** | 48.90 |
| | HIT@10 | 65.71 | 51.43 | **62.49** | 61.84 | 65.79 | 69.25 | 53.81 | 61.53 | 41.09 |
| 32 | MRR | 0.56 | **0.43** | 0.58 | 0.48 | 0.63 | 0.62 | 0.46 | 0.54 | 0.59 |
| | HIT@3 | 57.26 | **50.86** | 58.94 | 49.72 | 64.70 | 64.50 | 49.31 | 59.58 | 60.46 |
| | HIT@10 | 59.94 | **53.67** | 60.12 | 51.88 | 66.02 | 67.62 | 53.18 | **63.84** | 62.62 |
| 128 | MRR | - | - | - | - | - | - | - | - | **0.61** |
| | HIT@3 | - | - | - | - | - | - | - | - | **61.10** |
| | HIT@10 | - | - | - | - | - | - | - | - | **67.40** |

Table 6: Test results on FB122 dataset with different number of negative samples $n$ – we corrupt subject, entity, and both together, each with $n$ times, resulting in $3n$ total negative samples generated. Bold denotes column-wise best results. Due to the computational limit, we only evaluate CTP$_4$ when $n = 128$.

training procedure such as large number of sampling, along with careful hyper-parameter tuning, the model does not learn meaningful representations with larger and more complex datasets. This can be seen in Figure 8: while the embeddings learnt by pure KGE procedure (right) form clear clusters, the one obtained by CTP$_1$ (left) do not exhibit any structural pattern.

**Training NTPs with more negatives**  Given the above observation, we wish to see if the problem could be solved by training with more negatives, with results summarized in Table 6. Interestingly, instead of receiving better accuracy, we observe a drastic performance drop on CTP, CTP$_1$ when $\lambda = 0.8$ and CTP$_2$ with $\lambda = 0.5$ and $\lambda = 0.8$. For example, the MRR of CTP$_1$ with $\lambda = 0.8$ drops from 0.59 to 0.48 when number of negatives is increased from 1 to 32, and the MRR for CTP$_2$ with $\lambda = 0.8$ drops from 0.65 to 0.46. Reversely, when $\lambda = 0$, CTP$_1$'s MRR increases from 0.40 to 0.438 as number of negatives increases. This implies increasing the number of negatives helps when $\lambda$ is low, $i.e.$ when the KGE loss is contributing more to the gradient updates. However, even when $\lambda = 0$ for CTP$_1$, recovering a pure KGE optimization process, the accuracy with $n = 32$ is still far less than when $\lambda = 0.5$ and all other variants. This suggests that, while we show previously on small datasets that training with pure KGE objectives can offer a strong baseline for NTP inference, this phenomenon does not scale to larger and more complex datasets as in this case. A closer analysis on this scalability issue is required, which we will leave for future works. On the other hand, we notice drastic increase in accuracy with CTP$_4$ from 0.32 MRR to 0.61 with $n$ increases from 1 to 128.

## 6  CONCLUSION

In this paper we propose to leverage KGE methods to improve NTP performance by enhancing NTP's embedding space to be better structured and regularized. We explore four variants CTP$_1$ - CTP$_4$ for integrating KGEs into the NTP, and find that by injecting KGEs into NTP's score calculation (CTP$_2$) we can improve upon the baseline NTP by a large margin and achieve SOTA results on multiple datasets. We also show that we could drastically improve NTP's inference and evaluation performance by substituting computationally intensive NTP components with lightweight KGE operations. Finally, we conduct detailed ablations and analysis on key components of the integration.

**Limitation**  We also recognize limitations and directions for future work. First, while we show noticeable performance gain by integrating KGEs into NTPs, we also demonstrate in our ablations sections 5.3 and 5.3 that KGEs do not naturally reconcile with NTPs, where further analysis is required to examine the synergy between the two methods. Second, the efficiency of NTPs are still a concern. Although in CTP$_3$ and CTP$_4$ we reduce the inference and evaluation time drastically, it however comes with performance degradation, and is still lagging behind KGE methods. This hinders the usage of NTPs in real-world scenarios.

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

| Dataset | $|\mathcal{E}|$ | $|\mathcal{R}|$ | # Train | # Validation | # Test |
|---|---|---|---|---|---|
| Kinship (Kemp et al., 2006) | 104 | 25 | 8544 | 1068 | 1074 |
| Nations (Kemp et al., 2006) | 14 | 55 | 1592 | 199 | 201 |
| UMLS (Kemp et al., 2006) | 135 | 46 | 5,216 | 652 | 661 |
| FB122 (Guo et al., 2016a) | 9738 | 122 | 91,638 | 9595 | 11243 |
| WN18RR (Dettmers et al., 2018) | 40,943 | 11 | 86,835 | 3,034 | 3,134 |

Table 7: **Dataset statistics** Statistics of datasets used in this work. Columns: number of entities ($|\mathcal{E}|$), number of predicates ($|\mathcal{R}|$), number of training, validation, and test samples.

# 7 APPENDIX

## 7.1 DATASET INFORMATION

We conduct experiments on three small-scale link prediction datasets: Kinship, Nations and UMLS (Kemp et al., 2006), as well as two large-scale Knowledge Graph (KG) datasets: FB122 (Guo et al., 2016a) and WN18RR (Dettmers et al., 2018). FB122 is a subset of Freebase (Bollacker et al., 2007) containing facts of people, location and sports. Its test set is splitted into two subsets, Test-I and Test-II, where Test-I contains all triplets that *cannot* be derived by deductive logic inference, and Test-II denotes all the rest triplets. WN18RR is derived from WordNet (WN18) (Miller, 1995), where test triplets that can be obtained by inverting triplets in the training set are removed. In Table 7 we summarize the statistics of these datasets.

## 7.2 EXPERIMENTAL SETTINGS

**Rule templates** CTP defines a number of rule templates for the model to explore. The template is defined as number of steps – how many steps to hop from the head to the tail entity, and whether it is a reverse relation, indicated by $r$, $i.e.$ stepping from tail to head entity. For example, rule = 0 means the model will try to directly unify the goal with facts in the KB. rule = 2 means two steps from the head to the tail entity, $e.g.$ $R(s, o) :- R_1(s, z), R_2(z, o)$. rule = $1R$ means a reverse relation: $R(s, o) :- R_1(o, s)$. For Kinship, Nations and UMLS we follow the setting in CTP with Kinship=$\{0, 1, 1r\}$, Nations=$\{0, 2, 1r\}$, and UMLS=$\{0, 2\}$. For FB122 and WN18RR we both use $\{0, 1, 2, 1r\}$.

**Training** For hyper-parameters we follow CTP (Minervini et al., 2020) on Kinship and UMLS datasets for all the experiments. Specifically, we use embedding_size=50, top-$k$=4, batch_size=8, learning_rate=0.1, trained 100 epochs with Adagrad optimizer. For each triplets we sample 3 negative sample per entity (a total of 9 negative samples per triplet). For Nations we use batch_size=256 with AdamW optimizer for the $CTP_2$ variant, and the same as CTP for the rest of models. For FB122 and WN18RR we mostly follow the setting from GNTP (Minervini et al., 2019), with embedding_size=100, top-$k$=10, and 1 negative sample per entity. We use Adagrad optimizer and train 100 epochs.

For baseline CTP we find freezing the model entities in the first 25 epochs work well, while for all our CTP variants we receive better results by not freezing the model from the beginning. We also explore different score aggregation operations for aggregating scores along one proof path (AND operation). For baseline CTP and $CTP_1$ we find the original min generally work well, while mean and multiplication work better for $CTP_2$, $CTP_3$ and $CTP_4$. besides, we considered using cosine similarity as the scoring metric, with using addition instead of concatenation for obtaining the embedding for the whole triplets. However, we do not observe it to perform better than using the Gaussian kernel.

**Incorporating KGE objectives** To ensure KGE score lies within 0 and 1 we add a Sigmoid function to its negative score function. To avoid small negative scores being pushed to zeros after Sigmoid, we first subtract the mean from the negative scores.

| Kinship | Nations | UMLS | FB122 | WN18RR |
|---------|---------|------|-------|--------|
| *term21*(X,Y):-
*term24*(Y,Z)
*term4*(X,Y):-
*term4*(Y,Z)
*term9*(X,Y):-
*term11*(Y,Z) | *treaties*(X,Y):-
*treaties*(Y,X)
*aidenemy*(X,Y):-
*militaryactions*(Y,X)
*lostterritory*(X,Y):-
*timesincewar*(Y,X) | *associated_*with(X,Y):-
*process_*of(X,Y),
*process_*of(Y,Z)
*occurs_in*(X,Y):-
*issue_*in(X,Y),
*process_of*(Y,Z)
*interconnects*(X,Y):-
*result_of*(X,Y),
*result_of*(Y,Z) | *contains*(X,Y) :-
*capital*(Y, X)
*language_spoken*(X, Y):-
*official_language*(X,Y)
*place_lived*(X,Y):-
*place_of_birth*(X,Y) | *hypernym*(X,Y):-
*hypernym*(Y,X)
*verb_group*(X,Y):-
*verb_group*(Y,X)
*has_part*(X,Y) :-
*part_of*(Y,X) |

Table 8: Visualization of learnt rules under each dataset with CTP$_2$

## 7.3 VISUALIZATION OF LEARNT RULES

In Table 8 we show visualization results generated under each dataset under CTP2. We can see it successfully learns logical rules such as *place_lived*(X,Y):- *place_of_birth*(X,Y), *interconnects*(X,Y):- *result_of*(X,Y), *result_of*(Y,Z), and *contains*(X,Y) :- *capital*(Y, X).

## 7.4 PSEUDO-CODE IMPLEMENTATION

**Neural Theorem Prover** implements backward chaining algorithm by recursively instantiating AND/OR modules, where OR is called to prove each goal by unifying with each rule head in the KB. Then, the AND module is called to prove the rule body, where for each atom in the body the OR is recursively called, until the algorithm reaches depth limit $d$. The pseudo-code for NTP can be found in 1.

## 7.5 CONDITIONAL THEOREM PROVER

**Conditional Theorem Prover** extends upon NTP by incorporating a trainable neural module for predicting plausible rules given goals. The pseudo-code for CTP can be found in 2.

**Algorithm 1:** Python pseudo-code for NTP with top-$k$ retrieval following implementation from (Minervini et al., 2019)

```python
# KB: the Knowledge Base.

# S: proof state
#  - score: unification score
#  - subs: substitution set

# sim: similarity function for unification (Gaussian kernel)
# topk: a function that performs top-k retrieval

def or(goal, S, k):

    S_list = []
    for rule in KB:

        head, body = rule
        topk_ind = topk(goal, KB)

        if d < max_depth and no_cycle(S.subs, rule):

            S_head = unify(head, goal, S, topk_ind)
            S_head = kmax(goal, S_head)
            S_body = and(body, S, d)
            S_list.append(S_body)

    return proof_states

def and(goal, S):

    S_list = []
    if len(goal) == 0:
        S_list = [S]
    elif d < max_depth:

        goal, sub_goals = goal
        new_goal = substitute(goal, subs)

        for S_new in or(new_goal, S, d+1):
            S_list.append(and(sub_goals, S_new))

    return S_list

def unify(atom, goal, S, topk_ind):

    grounded_atom, grounded_goal = [], []
    for (atom_term, goal_term) in zip(atom, goal):

        if is_variable(atom_term):
            if atom_term not in S.subs: S.subs.update({atom_term: goal_term}
        elif is_variable(goal_term):
            if is_grounded(atom_term) and goal_term not in S.subs:
                S.subs.update({goal_term: atom_term}

        elif is_grounded(atom_term) and is_grounded(goal_term):
            grounded_atom.append(atom_term)
            grounded_goal.append(goal_term)

            score = sim(grounded_goal, grounded_atom[topk_ind])
            S.score = min(S.score, score)

    return S
```

**Algorithm 2:** Simplified Python pseudo-code for CTP following (Minervini et al., 2020)

```python
# KB: the Knowledge Base.
# sim: similarity function for unification (Gaussian kernel)
# topk: a function that performs top-k retrieval
# max_depth: maximum recursive depth

def ctp(s, r, o, max_depth):

    if max_depth == 0:
        return unify(s, r, o)
    else:
        score = None
        for d in range(max_depth):

            level_score = None
            for rule_path in rule_templates:

                path_score = None
                for step_ind, rule_transform in rule_path:

                    if is_inverse_relation:
                        latent_score, s = step(o, r, s, max_depth - 1)
                    else:
                        latent_score, s = step(s, r, o, max_depth - 1)

                    if path_score is None:
                        path_score = latent_score
                    else:
                        # min aggregation -- all proofs need to be hold.
                        path_score = min(path_score, latent_score)

                    if step_ind == len(rule_path):
                        # choose the max over the topk branches
                        path_score, _ = max(path_score, dim=-1)

                if level_score is None:
                    level_score = path_score
                else:
                    # max aggregation -- only one proof path needs to be hold.
                    level_score = max(level_score, path_score)

            if score is None:
                score = level_score
            else:
                # max aggregation -- only one proof path needs to be hold.
                score = max(score, level_score)

def unify(s, r, o=None):

    if o is not None:
        topk_ind = topk([s, r, o])
    else:
        topk_ind = topk([s, r])
        o = KB[topk_ind][-1]

    score = sim([s, r, o], KB[topk_ind])
    return score, o
```

