# OpenReview forum: "Improving Soft Unification with Knowledge Graph Embedding Methods"
_ICLR.cc/2025/Conference — Submitted to ICLR 2025_

### Official Review · Reviewer_iiQA · 2024-10-27

**Soundness:** 2
**Presentation:** 3
**Contribution:** 2
**Rating:** 5
**Confidence:** 3

**Summary:**

This paper proposes combining NTP with knowledge embedding methods to remedy NTP training difficulties. Specifically, the authors verified four combinations, and all of them show good performance across the symbolic reasoning datasets (Kinship, Nations, and UMLS). In some tasks, their methods give SOTA results. They also conducted some ablations experiments for configurations of KGEs.

**Strengths:**

1. This paper tries to solve a real problem of NTP: sparse proofs lead to training difficulties. The authors propose to use KGE to remedy this problem. By combining KGE and NTP, the embeddings become easier to train, because KGE is computationally dense and efficient.
2. The structure of the paper is clear, with four combinations to investigate: it proposes using KGE in various modules of NTP, including step, unification, and loss computation. The experimental results show the effectiveness of them.
3.  The experiments are extensive.

**Weaknesses:**

1. It claims that "We provide the first study for integrating KGEs into NTPs", however, it is not very appropriate. Although some integration methods proposed in this paper modify the NTP modules, the method that combines losses between NTP and KGE was already proposed in the first NTP paper [1]. In Sec. 4.2, it uses neural link prediction (ComplexE) as an auxiliary loss.
2. I believe that the equation in Line 108 is incorrect: the second term should be (1 - NTPxxx).
3. The proposed methods are trivial to some extent. Actually, except for the CTP1, the others seem to be irrelevant in easing training difficulties. Why does changing the unification equation improve training?
4. There are more systemic methods to convert the sparse computation into dense ones: [2] converts the grounding into dense matrix computation, [3] uses an Einsum operation for fast parallel groundings. They seem to be more promising in this direction.

[1]. End-to-End Differentiable Proving

[2]. Logic Tensor Network

[3]. LogicMP: A Neuro-symbolic Approach for Encoding First-order Logic Constraints

**Questions:**

Please see the weaknesses above.

---

> ### Author Response · Authors · 2024-11-19
>
> We thank the reviewer for the insightful comments. We are glad you find our work clear and extensive. We hope the following explanations will help address your concerns.
>
> ### *“the method that combines losses between NTP and KGE was already proposed in the first NTP paper”*
>
> We have changed “the first study” to “the first systematic study” in the updated manuscript. We are aware of the usage of ComplEx in the original NTP work and indeed mentioned it at L252-L254. However, we believe there are enough contents and findings that differentiate our work from the brief mention of KGE as auxiliary loss in the original NTP paper. Specifically:
> - As we mentioned in the paper (L252-L254), while the original NTP did include a section mentioning the use of ComplEx as an auxiliary loss, it did not include any further technical details or ablation studies, and such an approach is never used in the subsequent works [1,2,3]. Furthermore, as mentioend in [1] (page 6, footnote 7, bottom right), results in original NTP work were calculated with incorrect evaluation function which caused the resulting accuracy to be inflated.
> - On the other hand, in this work we aim at providing a systematic study of such integration. We show that using KGE as auxiliary loss ($\text{CTP}_1$) actually performs *worse* than baseline CTP on large-scale datasets (Table 3 and Table 4), where the other variant ($\text{CTP}_2$) performs noticeably better.
> - Besides focusing on accuracy, we also provide two other variants ($\text{CTP}_3$ and $\text{CTP}_4$) with a focus on training and evaluation efficiency.
> - Furthermore, we conduct extensive analysis on the key factors of the integration and analyze the integration from the embedding perspective.
>
> ### *“I believe that the equation in Line 108 is incorrect: the second term should be (1 - NTPxxx).”*
>
> We apologize for the mistake, and thank you for pointing out the error. We have corrected the formula in the updated manuscript.
>
> ### *“except for the CTP1, the others seem to be irrelevant in easing training difficulties. Why does changing the unification equation improve training?”*
>
> First, we would like to clarify that we are not only focusing on easing training difficulties ($\text{CTP}_1$ and $\text{CTP}_2$), but also efficiency ($\text{CTP}_3$ and $\text{CTP}_4$). The latter is not supposed to ease training difficulties.
>
> Second, we would like to clarify that in this work we do **not** view the training difficulties of NTPs from the sparse gradient perspective, but from the less-structured embedding perspective caused by the nature of the soft unification algorithm, as we mentioned in Section 3.4 (L186-L246). The rationale is: if there exists better global structure over the learnt embedding space of NTPs, then for two semantically-similar entities which have never been unified due to sparse proof, there are higher chances that they could have a higher similarity than other non-related entities. This differentiates ours from previous works [3, 4] that focuses directly on the sparse gradient problem. Both $\text{CTP}_1$ and $\text{CTP}_2$ do not improve the sparse proof problem, but they do improve the embedding space of NTPs to be better structured, as we show in Figure 2 and Figure 5.
>
> ### *“There are more systemic methods to convert the sparse computation into dense ones: Logic Tensor Network (LTN) and LogicMP. They seem to be more promising in this direction.”*
>
> First, as we mentioned above, while sparse proof is the real problem in NTPs, we take a different angle and tackle the problem from the embedding perspective.
>
> Second, LTN and LogicMP are very different neurosymbolic methods from NTPs. While they indeed do not have sparse computation problems as in NTPs due to different algorithm derivations, we do not see how to utilize them to improve the sparse proof problem that specifically resides in NTPs.
>
> The reason we choose to study NTPs is we believe the pure similarity-based unification mechanism is simple and more scalable than other neurosymbolic frameworks with more sophisticated tensor operations. Moreover, we believe it has the potential to be used in conjunction with the recent advances of foundational models, thereby making it multimodal with zero-shot capacity. This is indeed our next work.
>
> [1] Pasquale Minervini, Matko Bosnjak, Tim Rocktaschel, Sebastian Riedel, and Edward Grefenstette. “Differentiable reasoning on large knowledge bases and natural language” (AAAI 2020)
>
> [2] Pasquale Minervini, Sebastian Riedel, Pontus Stenetorp, Edward Grefenstette, and Tim Rocktaschel. “Learning reasoning strategies in end-to-end differentiable proving” (ICML 2020)
>
> [3] Jaron Maene and Luc De Raedt. “Soft-unification in deep probabilistic logic” (NeurIPS 2023)
>
> [4] Michiel de Jong and Fei Sha. Neural theorem provers do not learn rules without exploration

---

> > ### Comment · Reviewer_iiQA · 2024-11-22
> >
> > Thanks for your response. In Tables 2, 3, and 4, the performance of CTPx varies a lot. Which method should be used for different problems? As also mentioned by reviewer 6URk, how do you compare with the advanced KGE method, eg., NBFNet, in terms of performance?

---

> ### Author Response · Authors · 2024-11-22
>
> Thank you for your question! Please see the resposne below:
>
> ### *Which method should be used for different problems?*
>
> We recommend $\text{CTP}_2$ in general, as we observe $\text{CTP}_2$ generally performs the best among the four variants (L364). One special case where $\text{CTP}_1$ outperforms $\text{CTP}_2$ is on Kinship where the length of proof paths is limited to 1, making $\text{CTP}_2$ less effectiv (L367).
>
> ### *As also mentioned by reviewer 6URk, how do you compare with the advanced KGE method, eg., NBFNet, in terms of performance?*
>
> First, we would like to clarify that NBFNet is not a KGE model, but is GNN-based. For KGE-based models we are comparing with three popular KGE models: TransE, ComplEx and DistMult (Table 3 and 4).
>
> Below we show results with NBFNet included. For brevity we only include MRR here. We will add NBFNet to the table after we perform hyper-parameter search as NBFNet was not evaluated on the below datasets except WN18RR. We also include LERP, the other recent model that Reviewer 6URk mentioned.
>
> | Dataset    | Metric | NBFNet | LERP | CTP1  | CTP2     | CTP  | GNTP   | NeuralLP | MINERVA |
> |------------|--------|----------|-----|-----|----------|------|--------|----------|---------|
> |Kinship	  |MRR   | **0.80**   | 0.64 | 0.75| 0.71    |0.71 |0.719  |0.62|0.72 |
> | UMLS      | MRR    | 0.82   | 0.76 | 0.82  | **0.85** | 0.81 | 0.84   | 0.78     | 0.82    |
> | Nations    | MRR    | 0.75 | - |0.63   | **0.79** | 0.71 | 0.658  | -        | -       |
>
> | Dataset | Metric | NBFNet | LERP  | CTP1  | CTP2     | CTP  | GNTP   | ComplEx  | DistMult | NeuralLP | MINERVA |
> |---------|--------|----------|----|---|----------|------|--------|----------|----------|----------|---------|
> | FB122   | MRR    | **0.74**  | 0.66  | 0.56  | 0.68 | 0.63 | 0.61   | 0.64     | 0.63     | -        | -       |
> | WN18RR  | MRR    | 0.55| **0.62** | 0.36 | 0.42    | 0.36| 0.38  | 0.42    | 0.46    | 0.46    | 0.45   |
>
> We can see LERP performs exceptionally well on WN18RR, but is outperformed by $\text{CTP}_2$ on all other datasets by a large margin. On UMLS and Nations, $\text{CTP}_2$ performs better than NBFNet, while on large-scale dataset like FB122 and WN18RR, NBFNet outperforms CTPs by a large margin. However, we want to note that on one hand, given the relatively low baseline performance of NTPs, our goal is not to solely compete for the state-of-the-art, but rather to provide detailed analysis on the properties of NTPs. On the other hand, GNN-based models are known to generally hold better performance comparing to KGEs or neuro-symbolic methods, **at the cost of efficiency and interpretability**. Below we show the inference speed (in millisecond) under WN18RR dataset for some popular KGE models on a V100 GPU, a batch size of 8, embedding dimension 1000 for each KGE method (default), and CTP with dimension 100, and NBFNet with dimension=32. We can see NBFNet requires the longest inference time, $10 \times$ more than CTP. On the other hand, while CTP is slower than pure KGE-based models, we believe it is still a good compromise between speed and interpretability.
>
> | TransE | RotatE | ComplEx | DistMult | NBFNet | CTP |
> | --- | --- | --- | --- | --- | --- |
> | 9.3 | 40 | 0.32 | 0.24 | **113** | 8.2 |

---

> > ### Comment · Reviewer_iiQA · 2024-11-24
> >
> > Thank you for your reply. I will keep my original score.

---

> ### Author Response · Authors · 2024-11-25
> **Additional Experiments & Justifications**
>
> Below we show additional experiments on Countries, CoDEx-S on $\text{CTP}_2$ and NBFNet. We can see $\text{CTP}_2$ outperforms NBFNet on Countries, and performs on par with NBFNet on CoDEx-S, while outperforming recent neuro-symbolic method $\text{DiffLogic}^+$ [2].
>
> | Dataset  |  | Metric   | $\textbf{CTP}_2$  | CTP    | Neural LP | MINERVA | NBFNet | TransE | ComplEx | ConvE  | $\textbf{DiffLogic}^+$ [2] |
> |------------|----|------|--------|--------|-----------|---------|--------|--------|---------|--------|-----|
> | Countries  | S1       | AUC-PR | 100    | 100       | 100     | 100    | 100     | -       | -      | -|
> |            | S2       | AUC-PR | **94.23**  | 91.81     | 75.1    | 92.36  | 93.85  | -       | -      | -      |-|
> |            | S3       | AUC-PR | **96.5**   | 94.78     | 92.2    | 95.1   | 95.74  | -       | -      | -      |-|
> | CoDEx-S    |          | MRR    | 0.476  | 0.323     | 0.290       | -      | **0.483**  | 0.354   | 0.465  | 0.444   | 0.458 |
>
> Additionally, we also conduct a systematic generalization task on CLUTRR [1], where we train on graphs with 2, 3, and 4 edges, and test on graphs with 4, 6, 8, and 10 edges (hops). Results are summarized below. We can see $\text{CTP}_2$ outperforms baseline CTP by a noticeable margin, especially on $\text{Hops} = 8,10$, while outperforming NBFNet by a large margin, except when $\text{Hops} = 4$. This suggests CTP has much better systematic generation capacity, possibly due to its explicit rule-learning capacity, as compared to NBFNet.
>
> | Hops | $\textbf{CTP}_2$  | CTP    | NBFNet |
> |------|--------|--------|--------|
> | 4    | 0.991  | 0.978  | **0.998**  |
> | 6    | **0.983**  | 0.972  | 0.846  |
> | 8    | **0.977**  | 0.944  | 0.781  |
> | 10   | **0.958**  | 0.891  | 0.672  |
>
> We would like to summarize our view on the comparison between $\text{CTP}_2$ and GNN-based NBFNet:
> 1. **$\textbf{CTP}_2$ and NBFNet has their own advantages in case of performance:** $\text{CTP}_2$ generally performs better on smaller-scale statistical relational learning datasets, while NBFNet excels at large-scale KGs. Notably, $\text{CTP}_2$ outperforms NBFNet by a large margin on the systematic generalization task(CLUTRR) where the larger numbers of hops are unseen during training.
> 2. **$\textbf{CTP}_2$ has better interpretability:** while one can approximate the edges taken by the NBFNet to form an explanation, it lacks the logical interpretation, and it is hard to understand rules learnt by NBFNet, as compared to neuro-symbolic methods like $\text{NTPs}$.
> 3. **$\textbf{CTP}_2$ has better efficiency:** as we showed in our previous comment, NBFNet is much more computationally-expensive due to its adaptation of Bellman-Ford algorithm, and requires more than $10 \times$ time for inference as compared to $\text{CTP}$.
>
> We would also like to re-emphasize that given the relatively low baseline performance of NTPs, our contribution in this paper is not only to provide the state-of-the-art results across all methods, but also **1)** to provide a systematic study on the integration of NTPs and KGEs, **2)** to provide a novel view on the hardness of training NTPs from the perspective of the embedding space, and **3)** demonstrates that by integrating KGE objectives we can indeed improve such embedding space to be more structured, thus benefiting the performance.
>
> Lastly, we hope that the reviewer will kindly consider the merits of our work. If there are any additional concerns that we can address or discuss before the author-reviewer discussion period ends to raise your score, please let us know. Thank you again for your time and advise!
>
> [1] Koustuv Sinha, Shagun Sodhani, Jin Dong, Joelle Pineau, and William L. Hamilton. 2019. CLUTRR: A Diagnostic Benchmark for Inductive Reasoning from Text. (EMNLP 2019)
>
> [2] Shengyuan Chen, Huang Fang, Yunfeng Cai, Xiao Huang, and Mingming Sun. 2024. Differentiable neuro-symbolic reasoning on large-scale knowledge graphs. (NeurIPS 2023)

---

### Official Review · Reviewer_whZm · 2024-10-27

**Soundness:** 3
**Presentation:** 3
**Contribution:** 2
**Rating:** 6
**Confidence:** 4

**Summary:**

The paper proposes integrating Knowledge Graph Embedding (KGE) methods with Neural Theorem Provers (NTPs) to enhance neuro-symbolic reasoning systems, addressing the optimization challenges of NTPs. It presents four specific integration strategies (CTP1 to CTP4) and evaluates their performance across various datasets. The results demonstrate substantial improvements over baselines, especially in terms of Mean Reciprocal Rank (MRR) and inference speed, with CTP2 standing out as the most effective integration approach. Ablation studies and detailed analysis add depth to the findings, although limitations in scaling and efficiency are noted.

**Strengths:**

The integration of KGE methods with NTPs is well-motivated, bridging the optimization smoothness of KGE with the interpretability of symbolic logic inherent in NTPs. The systematic exploration of different integration strategies adds scientific rigor.

The paper evaluates the proposed methods across multiple benchmarks, showing a clear improvement over baselines. The use of detailed ablation studies, t-SNE visualizations, and analyses of embedding space provides strong empirical support for the proposed approach. Also, the authors applied 4 integration strategies (CTP1 to CTP4) and evaluates their performance across various datasets. The integration strategies are diverse and comprehensive.

The paper is well-organized and well-written. All 4 strategies are described with enough details.

**Weaknesses:**

While the proposed integration strategies are explained, some technical details, such as the choice of parameters for λ and specific KGEs for each variant, could be clarified further to assist replication and general understanding.

I am not familiar with the FB122 dataset. I know FB15K-237 is a commonly used KG reasoning database. The authors may want to explain why not using FB15K-237 but use FB122 instead.

Adding related work at the end is okay in general. But in this paper, the related work is entangled with the ablation study figures and tables, which makes the end of this paper hard to read. Please add related work after introduction.

**Questions:**

The authors may want to explain why not using FB15K-237 but use FB122 instead.

---

> ### Author Response · Authors · 2024-11-19
>
> Thank you for the constructive comments. We are glad you find our study thorough and well-organized. We hope the following explanations will help address your concerns.
>
> ### *“choice of parameters for λ and specific KGEs for each variant”*
>
> We apologize for the confusion. By default we use RotatE for $\text{CTP}_3$ and ComplEx for all other variants, because we observed empirically these configurations give the best overall performance. We use $\lambda = 0.5$ as default weight for combining NTP and KGE score/loss for all experiments except for Figure 6, where we conduct the ablation study on $\lambda$. We mentioned these specifications in the table captions in the original submission, but we have revised the Implementation section(L306-L307) and remove the redundant descriptions in captions in the updated manuscript to make this clear.
>
> ### *“FB122 vs. FB15K-237”*
>
> FB122 is a subset of FB15K-237 proposed in [1] and subsequently used in multiple works [2,3,4], including GNTP[4], a direct baseline of our method. It has two further subsets: Test-I and Test-II. Test-II denotes a subset of facts that can be inferred with logical rules, Test-I denotes all other facts. We use it instead of FB15K-237 because:
> - FB122 was adopted in GNTP, one of our direct baseline. By using FB122 it allows us to have a fair comparison without the need to re-perform hyper-parameter search on the baseline which may be suboptimal. Also, since the code of GNTP and most models that it compares to are written in tensorflow while we primarily use torch, we wouldn’t need to translate their source codes.
> - The NTP algorithms are still not as scalable as KGEs. We indeed tried FB15K-237 on baseline GNTP, however it took more than 70 hours on a V100 for both training and evaluation, which is far beyond the time and computation we could afford.
>
> ### *“the related work is entangled with the ablation study figures and tables ... Please add related work after introduction.”*
> We apologize for the bad formatting, and thank you for the suggestion. We have moved the related work section after introduction and updated the manuscript.
>
> [1] Guo, S.; Wang, Q.; Wang, L.; Wang, B.; and Guo, L. 2016. Jointly Embedding Knowledge Graphs and Logical Rules. (EMNLP 2016)
>
> [2] Minervini, P.; Demeester, T.; Rocktaschel, T.; and Riedel, S. 2017. Adversarial Sets for Regularising Neural Link Predictors (UAI 2017)
>
> [3] Garcıa-Duran, A., and Niepert, M. 2018. KBlrn: End-to-End Learning of Knowledge Base Representations with Latent, Relational, and Numerical Features. (UAI 2017)
>
> [4] Pasquale Minervini, Matko Bosnjak, Tim Rocktaschel, Sebastian Riedel, and Edward Grefenstette. “Differentiable reasoning on large knowledge bases and natural language” (AAAI 2020)

---

> ### Author Response · Authors · 2024-11-25
>
> Dear Reviewer whZm:
>
> We appreciate your valuable comments on our paper. We have prepared a rebuttal with an updated manuscript and tried our best to address your concerns. We notice that the author-reviewer discussion period is coming to an end, and we are willing to answer any unresolved or further questions that you may have regarding our rebuttal if time is allowed.
>
> If our rebuttal has addressed your concerns, we would appreciate it if you would let us know of your final thoughts. Additionally, we will be happy to answer any further questions regarding the paper. Thank you for your time and consideration.

---

### Official Review · Reviewer_5dZR · 2024-11-04

**Soundness:** 3
**Presentation:** 2
**Contribution:** 3
**Rating:** 6
**Confidence:** 4

**Summary:**

This work proposes to integrate a knowledge-graph embedding (KGE) loss function into the optimization of a neural theorem prover (NTP) model, a type of neurosymbolic search architecture that is particularly hard to optimize due to subgradient sparsity in its backward pass. The goal of this integration is to alleviate the optimization challenges faced by the standard NTP objective, which contains many operations with sparse gradients and is plagued by local minima. The authors explore several alternative formulations of this integration,
all based on the Conditional Theorem Prover (CTP; Minervini et al., 2020):
- CTP$_1$: a straightforward linear mixture of the original NTP loss and the KGE loss
- CTP$_2$: adding the KGE scoring function to the NTP stepwise unification scores
- CTP$_3$: changing the goal-reformulation module of the CTP to use the KGE's tuple completion geometry, viable for path-based KGE methods that support calculating the tail entity embedding in a triple directly from the head entity and relation embeddings
- CTP$_4$: replacing the final unification computation in a CTP proof path (normally a very large kernel evaluation) with a KGE lookup.

The authors evaluate their approaches on several link prediction tasks against three prior NTP variants, various KGE methods, Minerva, and NeuralLP. On the Kinship, Nations, UMLS, and FB122 (Test-II and Test-ALL splits), one of CTP$_1$ or CTP$_2$ performs best. On the FB122 Test-I split and WN18RR, KGE baselines outperform the proposed systems.

The authors additionally include analysis of the learned embedding spaces and training dynamics, including the observation that on large datasets like WN18RR, the embedding spaces learned by their hybrid methods do not appear to recover the structure seen in the spaces learned by the baseline KGE methods.

**Strengths:**

- I feel that the authors successfully demonstrate that hybridizing the two approaches (KGE and NTP) can yield complementary benefits.
- The analysis and ablations are very thorough: I found that they improved my understanding of the differences between the proposed systems and fairly characterized the remaining shortcomings of the best-performing options.
- The paper does a good job of describing the NTP background clearly. While I'm familiar with the method, it doesn't seem like it would be hard to follow for someone who was less so.

**Weaknesses:**

- While the proposed methods are successful on certain datasets, they fail to match baselines in certain cases (as the authors note). The paper presents clear diagnostic evidence of this issue in section 4.3 and some speculation on why this occurs, but the fact remains that the proposed methods are clearly situational and not straightforwardly scalable to large/complex domains.
- The specifics of CTP$_2$, CTP$_3$ and CTP$_4$ could use more elaboration; it's unclear how the KGE negative samples are included in these cases.

**Questions:**

Q1: In CTP$_2$, CTP$_3$, and CTP$_4$, how are contrastive negatives included? Is the original KGE objective still optimized (it seems like it's not)?

---

> ### Author Response · Authors · 2024-11-19
>
> Thank you for the helpful and constructive comments. We are glad you find our study thorough, helpful. We hope the following explanations will help address your concerns.
>
> ### *“the proposed methods are clearly situational and not straightforwardly scalable to large/complex domains.”*
>
> **Compared to baseline NTPs**, while $\text{CTP}_1$ is situational which only performs well on small-scale datasets and does not scale to large datasets (FB122 and WN18RR),  we find that $\text{CTP}_2$ generally outperforms baseline NTPs by a large margin across both small-scale and large-scale datasets.
>
> **Compared to state-of-the-art KGE models**, we agree with the reviewer that $\text{CTP}_2$ is behind the state-of-the-art KGE on WN18RR(Table 4). However, it still shows large improvements against baseline NTPs. On the other hand, on FB122 (Table 3), $\text{CTP}_2$ outperforms both NTP baselines and KGE methods by a noticeable margin. We believe these show $\text{CTP}_2$ is not purely situational and can scale to large-scale dataset, and is a good compromise between the performance and scalability of pure latent models like KGEs, and neuro-symbolic NTP framework.
>
> ### *“the specifics of CTP , CTP and CTP could use more elaboration; it's unclear how the KGE negative samples are included in these cases.”*
>
> We apologize for the confusion. The negative samples are included in the same way in all cases: for each triplet, we corrupt subject, entity, and both together, each with n times, resulting in $3n$ total negative samples generated per triplet. These negative samples will receive negative label $y = 0$, and the model is trained according to the NTP objective (Equation 1). We have revised the Implementation section(L305-L306) to include the description in the updated manuscript.
>
> ### *“is the original KGE objective still optimized (it seems like it's not)?”*
>
> We apologize for the confusion. We do not optimize the KGE objective for $\text{CTP}_{2-4}$. We have revised the description in the method section(L227-L228) to make this clear in the updated manuscript.

---

> ### Author Response · Authors · 2024-11-25
>
> Dear Reviewer 5dZR:
>
> We appreciate your valuable comments on our paper. We have prepared a rebuttal with an updated manuscript and tried our best to address your concerns. We notice that the author-reviewer discussion period is coming to an end, and we are willing to answer any unresolved or further questions that you may have regarding our rebuttal if time is allowed.
>
> If our rebuttal has addressed your concerns, we would appreciate it if you would let us know of your final thoughts. Additionally, we will be happy to answer any further questions regarding the paper. Thank you for your time and consideration.

---

> ### Comment · Reviewer_5dZR · 2024-12-02
>
> Thanks for addressing my questions! I'm keeping my score the same.

---

### Official Review · Reviewer_6URk · 2024-11-04

**Soundness:** 3
**Presentation:** 3
**Contribution:** 2
**Rating:** 6
**Confidence:** 3

**Summary:**

This paper handles the link prediction by combining Neural Theorem Proving (NTP) and Knowledge Graph Embeddings (KGE). Specifically, the KGE is injected into the CTP framework (a successor of NTP) in four ways. Empirical studies on large-scale datasets justified the empirical performances of those four ways of integration. Notably, CTP 1 and 2 perform way better than but significant slower than CTP 3 and 4.

**Strengths:**

- The empirical study is systematic and details the effectiveness of four variants and the impact of some crucial hyperparameters, including the weight and number of negative samples.
- the visualization also helped people understand the role of KGEs, which provides better global embedding structures.

**Weaknesses:**

- One of the key weaknesses is that all findings developed in this paper are solely for neural theorem provers. This makes the paper less impactful, given the existence of other technical solutions, such as GNN-based link predictors like NBFNet, rule miners like LERP, and a large language model for knowledge harvesting.
- In light of the weakness above, it would be much better if the authors could also analyze the proof produced when conducting link predictions as the key feature of neural link predictors.

**Questions:**

- In terms of performance, what is the performance and speed of using KGE to make link predictions?

---

> ### Author Response · Authors · 2024-11-19
>
> Thank you for the insightful comments. We are glad you find our study systematic. We hope the following explanations will help address your concerns.
>
> ### *"focusing only on NTP given the existence of other technical solutions such as GNN-based link predictors like NBFNet, rule miners like LERP, and a large language model for knowledge harvesting”*
>
> We agree there are numerous approaches for link prediction. We choose to focus on the integration of Neural Theorem Provers (NTPs) and Knowledge Graph Embedding (KGE) methods because they are both embedding-based methods, where the similarity-based unification operation and the objective formulation in NTPs share interesting similarity with KGEs (L181), and their properties complement each other. Moreover, because of NTPs’ pure similarity-based unification operation, we believe it has the potential to be used in conjunction with the recent advances of foundational models, thereby making it multimodal with zero-shot capacity. This is indeed our next work.
>
> On the other hand, while we acknowledge the diversity of available methods, we believe it is neither necessary nor feasible for a single work to cover findings relevant to all paradigms. Meanwhile, NTP represents a general neuro-symbolic **framework** derived upon backward chaining, rather than a **specific** algorithm, and several prior works [1, 2, 3] have also focused solely on the NTP framework.
>
> Regarding the specific methods mentioned by the reviewer:
> - *NBFNet (NeurIPS 2021)* is a GNN-based link predictor that parametrizes the Bellman-Ford algorithm using neural components. This approach is fundamentally distinct from both NTPs and KGEs, and we believe it would be challenging to integrate into the current framework without deviating from our focus.
> - *LERP (ICLR 2023)* proposes a method to encode local subgraphs as additional context for each entity in the knowledge base. We do see the possibility of using it as an alternative for KGE under the NTP framework. However, as it is a recent work, we were unaware of it during our study. Also, since it is quite distinct from KGE methods, we are not sure if we should include it in parallel with other KGE methods in a single work. Therefore, we will leave it for future work.
>
>
> ### *"analyze the proof produced when conducting link predictions"*
> We appreciate the reviewer’s suggestion to analyze and visualize the proofs produced during link prediction. In our original submission, we did not include these visualizations because our work does not propose a new algorithm but rather focuses on the integration of NTPs and KGEs. Consequently, we prioritized visualizations of the learned embeddings, which we believed offered more direct insight into our contributions.
>
> However, as per the reviewer’s suggestion, we have visualized the proofs generated by the model for each dataset under CTP2 as shown below. We have also included them in Table 8 in the updated manuscript.
>
> | Kinship | Nations | UMLS | FB122 | WN18RR |
> | -------- | --------- | ------- | -------- | ----------- |
> | term21(X,Y):- term24(Y,Z) | treaties(X,Y):- treaties(Y,X) |associated_with(X,Y):- process_of(X,Y), process_of(Y,Z) | contains(X,Y) :- capital(Y, X)  | hypernym(X,Y):- hypernym(Y,X) |
> | term4(X,Y):- term4(Y,Z) | aidenemy(X,Y):- militaryactions(Y,X) | occurs_in(X,Y):- issue_in(X,Y), process_of(Y,Z)    | language_spoken(X, Y):- official_language(X,Y) | verb_group(X,Y):- verb_group(Y,X) |
> | term9(X,Y):- term11(Y,Z) | lostterritory(X,Y):- timesincewar(Y,X) |interconnects(X,Y):- result_of(X,Y), result_of(Y,Z)  | place_lived(X,Y):- place_of_birth(X,Y) | has_part(X,Y) :- part_of(Y,X) |
>
> From the above table, we can see the model successfully learns logical rules such as place_lived(X,Y):- place_of_birth(X,Y), interconnects(X,Y):- result_of(X,Y), result_of(Y,Z), and contains(X,Y) :- capital(Y, X).
>
> ### *"in terms of performance, what is the performance and speed of using KGE to make link predictions?"*
>
> Below we show the inference speed under WN18RR dataset for some popular KGE models on a V100 GPU, a batch size of 8, embedding dimension 1000 for each KGE method (default), and CTP with dimension 100. In general KGE models enjoy fast training/inference speed due to its relatively simple tensor operations. For accuracy for KGE methods, please refer to Table 3 and Table 4 in the paper.
>
> | TransE | RotatE | ComplEx | DistMult | CTP |
> | --- | --- | --- | --- | --- |
> | 9.3 | 40 | 0.32 | 0.24 | 8.2 |
>
> [1] Pasquale Minervini, Matko Bosnjak, Tim Rocktaschel, Sebastian Riedel, and Edward Grefenstette. “Differentiable reasoning on large knowledge bases and natural language” (AAAI 2020)
>
> [2] Pasquale Minervini, Sebastian Riedel, Pontus Stenetorp, Edward Grefenstette, and Tim Rocktaschel. “Learning reasoning strategies in end-to-end differentiable proving” (ICML 2020)
>
> [3] Jaron Maene and Luc De Raedt. “Soft-unification in deep probabilistic logic” (NeurIPS 2023)

---

> ### Author Response · Authors · 2024-11-25
>
> Dear Reviewer 6URk:
>
> We appreciate your valuable comments on our paper. We have prepared a rebuttal with an updated manuscript and tried our best to address your concerns. We notice that the author-reviewer discussion period is coming to an end, and we are willing to answer any unresolved or further questions that you may have regarding our rebuttal if time is allowed.
>
> If our rebuttal has addressed your concerns, we would appreciate it if you would let us know of your final thoughts. Additionally, we will be happy to answer any further questions regarding the paper. Thank you for your time and consideration.

---

> ### Author Response · Authors · 2024-11-27
> **Additional experiments & comparison between NTPs and NBFNet**
>
> Dear Reviewer 6URk,
>
> We just want to bring your attention to our responses to iiQA's question, where we conduct more experiments and comparison between $\text{NTP}$s and NBFNet. To summarize:
>
> 1. **$\textbf{CTP}_2$ and NBFNet has their own advantages in case of performance:** $\text{CTP}_2$ generally performs better on smaller-scale statistical relational learning datasets, while NBFNet excels at large-scale KGs. Notably, $\text{CTP}_2$ outperforms NBFNet by a large margin on the systematic generalization task(CLUTRR) where the larger numbers of hops are unseen during training.
> 2. **$\textbf{CTP}_2$ has better interpretability:** while one can approximate the edges taken by the NBFNet to form an explanation, it lacks the logical interpretation, and it is hard to understand rules learnt by NBFNet, as compared to neuro-symbolic methods like $\text{NTPs}$.
> 3. **$\textbf{CTP}_2$ has better efficiency:** as we showed in our previous comment, NBFNet is much more computationally-expensive due to its adaptation of Bellman-Ford algorithm, and requires more than $10 \times$ time for inference as compared to $\text{CTP}$.
>
> Let us know if our rebuttal has addressed your concerns, and please let us know if you have any further questions regarding the paper. Thank you for your time and consideration.

---

> > ### Comment · Reviewer_6URk · 2024-12-03
> > **Dear authors, thanks for your feedback.**
> >
> > I think my concerns have been addressed. I decided to give this work a 6 to show my support for valid neuro-symbolic reasoning.

---

### Author Response · Authors · 2024-11-19
**General Response**

Dear Reviewers, Area Chairs, and Program Chairs,

We thank all the reviewers for their time and their thoughtful feedback. We are encouraged that they find our work well-motivated (Rev. whZm),  analysis rigorous (Rev. whZm),  thorough and extensive (Rev. 6URk, 5dZR, whZm, iiQA), clearly-written and easy to follow (5dZR, whZm, iiQA). Below we summarize the main contribution of this paper:
- We provide the first systematic study for integrating Knowledge Graph Embedding(KGE) methods into Neural Theorem Provers (NTP) and propose four integration strategies, with two focusing on performance, and the other two on efficiency.
- We show that the integration noticeably improve the baseline NTP by a large margin and achieve SOTA results on multiple datasets. We also show that by leveraging the properties of KGEs we could drastically improve the inference and evaluation efficiency.
- We provide detailed ablations to examine the learnt embedding of NTPs and key factors in the integration. We also provide the first analysis on the embedding space of NTPs, and show that while the embedding space of original NTPs can be poorly structured due to the sparse proofs and the nature of soft unification, we can improve it to be better structured by incorporating KGE score/loss into NTPs, thus improving accuracy. Interestingly, we also find NTPs can achieve superior results with pure KGE objectives under several datasets, suggesting the synergy between the two distinct methods.

Given the reviewers' constructive feedbacks, we made the following changes to our paper:
- We add a visualization of learnt rules over all datasets (Table 8 in Appendix) (Rev. 6URk)
- We add a more detailed description of the procedure of including negative samples in the Implementation Section (L304-L306). (Rev. 5dZR)
- We add a more detailed description regarding when the original KGE objective is optimized in the Method section(L227-L228). (Rev. 5dZR)
- We move the Related Work section at the end of the paper to be after the Introduction section. (Rev. whZm)

Here we address a common concern from reviewer 6URk and iiQA:

*Why focusing on NTPs while there are other alternative neural link predictors?*

The reason we choose to study NTPs is we believe the pure similarity-based unification mechanism is simple and more scalable than other neurosymbolic frameworks with more sophisticated tensor operations. Moreover, we believe it has the potential to be used in conjunction with the recent advances of foundational models, thereby making it multimodal with potential zero-shot capacity. This is indeed our next work. On the other hand, this also emphasizes the importance of our analysis on the embedding space of NTPs.

Again, we thank all the reviewers for the elaborate comments and spending time to review the paper. Please let us know if you have any further concerns or recommendations for  improving the paper.

Best regards,

Authors of Paper 3869

---

### Meta-Review · Area_Chair_etEG · 2024-12-07

**Metareview:**

This paper systematically explores methods for combining knowledge graph embeddings with a neural theorem prover, specifically the Conditional Theorem Prover (CTP). Knowledge graph embeddings are combined with the CTP objective in 4 different ways.  CTP1 is an auxiliary loss as discussed in the original NTP. CTP2 ensembles scores from the CTP and KGE.  CTP3 and CTP4 are focused on improving efficiency using KGE.

The paper presents results on several link prediction tasks. CTP2, the strongest method, mostly outperforms the baselines.  On some datasets, it only outperforms ComplEx by a bit. For instance, on WN18RR, there are gains in H@1 but not @3 or @10.

This paper covers interesting ground in terms of improving methods for training neural theorem provers. The idea of auxiliary objectives to improve this training is attractive. The empirical results in the paper are very thorough.

I think the main drawback of this paper is the lack of a strong takeaway message, partially because four different model variants are explored.  As we dig into the results, I think the contribution may be a bit narrower than the paper frames it.

- As 6URk points out, there are other methods for neural link prediction. CTP2 and NBFNet have complementary strengths (as described in the discussion period), so these advances will only impact certain problems. The paper is up front about this and the contributions to NTPs are nevertheless valuable!

- Integration of KGEs has already been explored in the original NTP paper (as iiQA points out). So what we expect to see here is a method that's really impactful for NTPs specifically, since the impact will be somewhat limited to NTPs themselves.

- The ultimate best-performing method, CTP2, ensembles scores from the NTP with KGE. Looking at FB122, this is able to combine the strengths of CTP with ComplEx, but it doesn't seem transformative, more like a combination of these two methods.

This, plus the broader focus on CTP3 and CTP4 as well, gives this paper value as a systematic study, but also makes it harder to find a strong, exciting, novel takeaway message.

**Additional Comments On Reviewer Discussion:**

The authors clarified several points during the discussion period. Beyond minor details and clarifications, there are a few more major points:

- Comparison with NBFNet, another link prediction method. These are very helpful results for contextualizing the discussion in the paper and situating its contributions with respect to other methods.

- Analyzing proofs (6URk) and visualizing learned rules

- Clarifying which method is most recommended (discussion with iiQA). This is helpful and I think can tighten the paper with a more concrete goal.

However, while these points are valuable, they do not change the fundamental assessment I've described above.

---

### Decision · Program_Chairs · 2025-01-22

Reject